# A facile dual-template-directed successive assembly approach to hollow multi-shell mesoporous metal–organic framework particles

Haidong Xu ⬡[1], Ji Han[1], Bin Zhao[1], Ruigang Sun[1], Guiyuan Zhong[1], Guangrui Chen[1], Yusuke Yamauchi ⬡[2,3] & Buyuan Guan ⬡[1,4] ✉

Hollow multi-shell mesoporous metal–organic framework (MOF) particles with accessible compartmentalization environments, plentiful heterogeneous interfaces, and abundant framework diversity are expected to hold great potential for catalysis, energy conversion, and biotechnology. However, their synthetic methodology has not yet been established. In this work, a facile dual-template-directed successive assembly approach has been developed for the preparation of monodisperse hollow multi-shell mesoporous MOF (UiO-66-NH$_2$) particles through one-step selective etching of successively grown multi-layer MOFs with alternating two types of mesostructured layers. This strategy enables the preparation of hollow multi-shell mesoporous UiO-66-NH$_2$ nanostructures with controllable shell numbers, accessible mesochannels, large pore volume, tunable shell thickness and chamber sizes. The methodology relies on creating multiple alternating layers of two different mesostructured MOFs via dual-template-directed successive assembly and their difference in framework stability upon chemical etching. Benefiting from the highly accessible Lewis acidic sites and the accumulation of reactants within the multi-compartment architecture, the resultant hollow multi-shell mesoporous UiO-66-NH$_2$ particles exhibit enhanced catalytic activity for CO$_2$ cycloaddition reaction. The dual-template-directed successive assembly strategy paves the way toward the rational construction of elaborate hierarchical MOF nanoarchitectures with specific physical and chemical features for different applications.

MOFs have received widespread interest in various fields because of their high diversity in chemical composition, pore architecture, and macroscopic morphology[1–4]. The rational design and construction of MOF nanoarchitectures with higher complexity and more elaborate functionality have become one of the focus research areas in MOF growth[5–9]. In recent decade, hollow multi-shelled structures (HoMSs) have been emerging and developing rapidly as an important class of functional materials for catalysis, energy, and biomedicine-related applications[10–22], because of their unique compartmentation environments, abundant heterogeneous interfaces, and diverse available

[1]State Key Laboratory of Inorganic Synthesis and Preparative Chemistry, College of Chemistry, Jilin University, Qianjin Street 2699, Changchun 130012, PR China. [2]School of Chemical Engineering and Australian Institute for Bioengineering and Nanotechnology (AIBN), The University of Queensland, Brisbane, QLD 4072, Australia. [3]Department of Materials Process Engineering, Graduate School of Engineering, Nagoya University, Furo-cho, Chikusa-ku, Nagoya, Aichi 464-8603, Japan. [4]International Center of Future Science, Jilin University, Qianjin Street 2699, Changchun 130012, PR China. ✉e-mail: guanbuyuan@jlu.edu.cn

chemical compositions. Many interesting phenomena and enhanced performances originate from diverse HoMSs constructed in purposeful ways[23–29]. Recently, MOF (MIL-101) HoMSs have been successfully prepared and shown enhanced catalytic activity in styrene oxidation reaction, because the accumulation capability of the reactants in the compartments of hollow multi-shell MOF nanostructures enhances the catalytic kinetics[30]. However, the microporous texture of conventional MOFs imposes great obstacles on mass diffusion of the reactants through the pore systems, which is difficult to make full utilization of the catalytically active sites inside MOF particles. Therefore, an approach that could achieve hollow multi-shell MOF particles with controllable mesochannels (>5 nm) in highly open configuration may greatly extend applicability of MOFs to heterogeneous catalysis[31–34].

The selective etching of multi-layer materials with inhomogeneous layers, in which the outer part of each layer is chemically more robust than the inner part, is one of the most effective approaches to prepare multi-shell particles[35–37]. Therefore, engineering MOF layers with different chemical stability between their inner and outer parts is critically important for selective etching. Recently, mesoporous MOF particles have been prepared via the cooperative self-assembly of amphiphilic molecular templates and MOF precursors[38–41]. However, in these cases, all of the resultant mesoporous MOFs exhibit homogeneous frameworks, because it is unusual to involve two or more distinct growth stages in one assembly process between single soft templates and MOF precursors to form mesostructures with well-controlled inhomogeneity. Moreover, it is still difficult to rationally assemble inhomogeneous mesostructured MOF layers in desirable ways and let alone intelligently selective etching of chemically unstable parts in inhomogeneous MOF layers to produce multi-shell hollow MOF particles with highly accessible mesopores.

Herein, we demonstrate a facile dual-template-directed successive assembly strategy to synthesize mesoporous UiO-66-NH$_2$ hollow multi-shell particles (denoted as nS-mesoUiO-66-NH$_2$ HoMSs, where n indicates the number of shells). The innovative dual-template-directed self-assembly enables the synthesis of inhomogeneous UiO-66-NH$_2$ particles, featuring a less stable mesoporous core and a more stable mesoporous shell. Then, repetitive dual-template-directed assembly processes allow the formation of MOF precursor particles with multiple inhomogeneous layers consisting of two different mesostructured MOFs. At last, the region-specific etching is performed to generate multi-shell hollow MOF particles with highly open mesochannels (Fig. 1a). The mesoUiO-66-NH$_2$ HoMSs possess controllable shell numbers up to seven layers, accessible mesochannels (7.6 nm), large pore volume (0.64 cm$^3$ g$^{-1}$), tunable shell thickness and chamber sizes. Moreover, this method can also be used to synthesize other mesoporous MOF HoMSs. Compared with conventional microporous MOF crystals, the mesoUiO-66-NH$_2$ HoMSs with accessible Lewis acidic sites and enhanced accumulation capacity for the reactants exhibit enhanced catalytic activity and good stability in the CO$_2$ cycloaddition reaction with epoxides. The proposed synthetic method and the hereunto unfound influence of multiple templates on sequence assembly behaviors show great potential for the creation of advanced hierarchical MOF nanostructures with high architectural diversity and elaborate functionality.

## Results and discussion

Figure 1b and Supplementary Fig. 1a shows an overview scanning electron microscopy (SEM) image of the as-prepared 3S-mesoUiO-66-NH$_2$ HoMSs. The MOF particles show uniform round-edge polyhedral shape with an average diameter of ~400 nm. Closer observation of a multi-shell particle at a higher magnification reveals that dense mesopores are exposed on its surface (Fig. 1c). Furthermore, the multi-shell hollow structure can be well identified from a broken particle (Fig. 1d). As further confirmed by the transmission electron microscopy (TEM) observation (Fig. 1e and Supplementary Fig. 1b), all 3S-

mesoUiO-66-NH$_2$ HoMSs possess a triple-layer hollow structure. The inner, intermediate, and outer shells of a triple-shell particle show the diameters of 140, 270, and 410 nm and the shell thicknesses of 50, 50, and 55 nm, respectively (Fig. 1f). Besides, the mesochannels in each MOF shell are arranged radially, and the pore size is estimated to be ~7 nm (Fig. 1g). The highly open pore architecture could favor the mass transport of molecules throughout the MOF nanostructure. EDX elemental mapping performed on the 3S-mesoUiO-66-NH$_2$ HoMSs shows that the Zr, C, and N elements are uniformly distributed in the three-layer nanoarchitecture (Fig. 1h).

Powder X-ray diffraction (XRD) analysis confirms the crystallographic structure and phase purity of the resultant 3S-mesoUiO-66-NH$_2$ HoMSs. The sharp and intense diffraction peaks of 3S-mesoUiO-66-NH$_2$ HoMSs match well with those of microporous UiO-66-NH$_2$ crystals (Fig. 1i and Supplementary Fig. 2), indicating the same crystalline structure of two MOF samples. Distinct from the microporous nature of UiO-66-NH$_2$ nanocrystals, 3S-mesoUiO-66-NH$_2$ sample shows two major capillary condensation steps in the relative pressure ranges of 0−0.05 and 0.60−0.85 in the N$_2$ sorption isotherms (Fig. 1j), implying the coexistence of micropores and mesopores in MOF matrix. The specific surface area and total pore volume of 3S-mesoUiO-66-NH$_2$ HoMSs are calculated to be 812 m$^2$ g$^{-1}$ and 0.64 cm$^3$ g$^{-1}$, respectively. The Barrett-Joyner-Halenda (BJH) mesopore size distribution (Fig. 1k) further reveals the average mesopore size is ~7.6 nm. The thermogravimetric (TG) analysis indicates that both the 3S-mesoUiO-66-NH$_2$ HoMSs and microporous UiO-66-NH$_2$ crystals exhibit similar thermal stability (Supplementary Fig. 3).

The formation of 3S-mesoUiO-66-NH$_2$ HoMSs is mainly based on the creation of MOF precursor particles consisting of multiple inhomogeneous layers with alternating two different mesostructured MOFs and selective etching to remove the chemically unstable layers. It should be noted that two types of amphiphilic molecular templates, ionic surfactant octadecyl dimethyl betaine (ODMB) and polymer surfactant F127, play key roles in manipulating the cooperative self-assembly kinetics for successive epitaxial growth of two mesostructured MOF layers and their relative stability upon chemical etching with acetic acid. In order to better understand the effect of the repetitive dual-template-directed epitaxial growth and selective etching towards the formation of multi-shell mesoporous MOF nanostructure, a series of control experiments have been carried out. Figure 2a schematically illustrates the structure of different UiO-66-NH$_2$ nanoparticles. First, the experiments using single soft template are carried out to probe their assembly behaviors and framework stability upon chemical etching. The UiO-66-NH$_2$ particle prepared with ionic surfactant ODMB (Supplementary Fig. 4) as sole soft template shows a core-shell structure with disordered worm-like mesochannels in the core and microporous framework in the shell (Fig. 2b and Supplementary Figs. 5 and 6). The mesoporous core is formed by fast assembly of ODMB and MOF precursors at the initial stage of the reaction. The microporous MOF layer epitaxially grows on the mesoporous core after the depletion of soft templates. A following-up etching experiment is carried out to test the stability of the mesoporous core in the obtained MOF nanostructure in acetic acid solution. The central part of each MOF particle is selectively etched away to produce a hollow architecture (Supplementary Fig. 7). The higher density of defects in the central region may cause its faster etching in acidic media. For the UiO-66-NH$_2$ particles synthesized with block copolymer F127 as sole soft template, uniform nanocrystals are formed with mesopores evenly distributed throughout the whole particle (Fig. 2c and Supplementary Figs. 8 and 9). Compared with the mesostructure prepared with ODMB template, the resultant MOF particles with thicker mesopore walls show stronger resistance to the chemical etching in acetic acid solution (Supplementary Fig. 10). Second, the dual-template-directed cooperative assembly is studied through one-step co-assembly of both ODMB and F127 with MOF

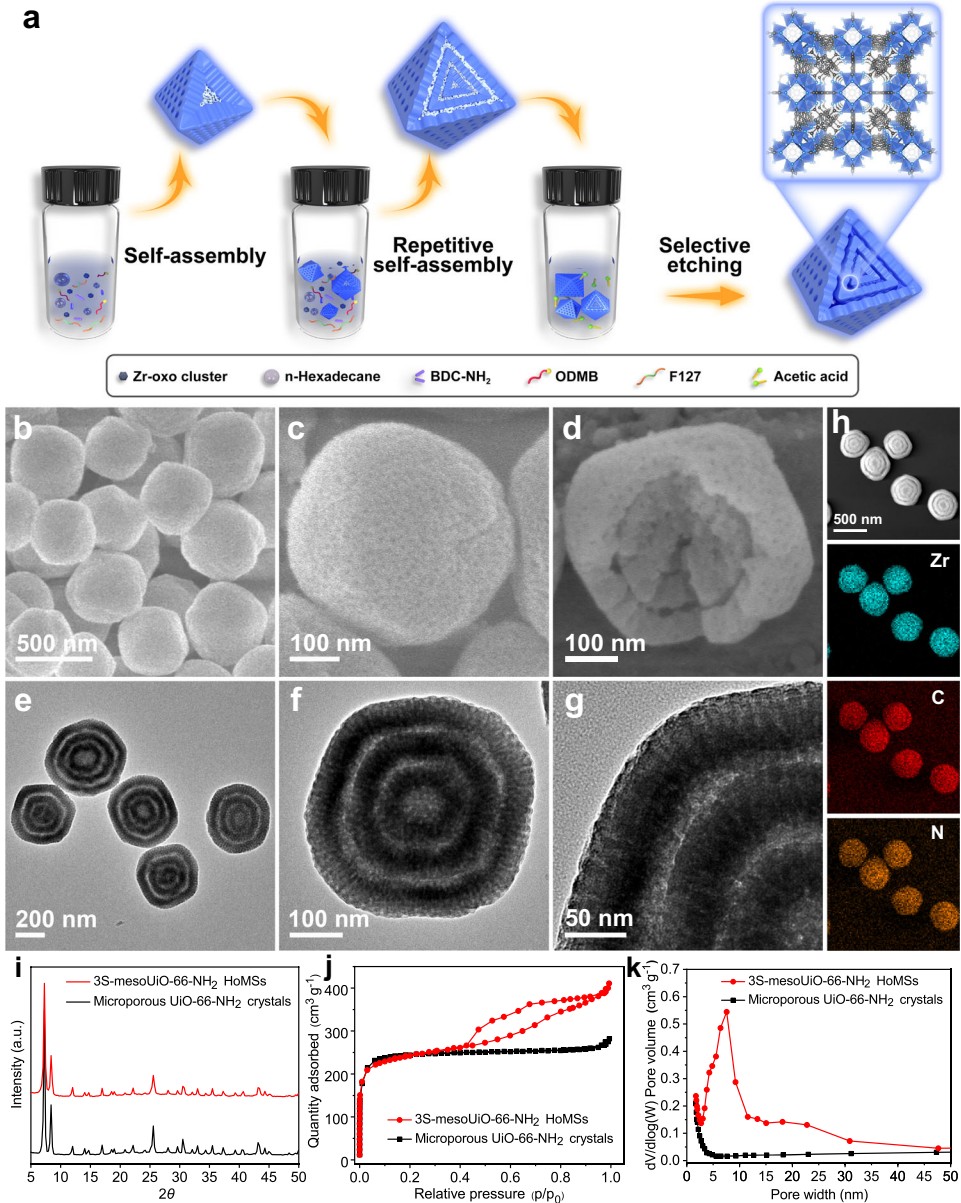

**Fig. 1 | Synthesis and characterization of the 3S-mesoUiO-66-NH₂ HoMSs.** **a** Schematic illustration of the synthesis process. **b**–**d** SEM and **e**, **f** TEM images of 3S-mesoUiO-66-NH₂ HoMSs. **g** Magnified TEM image showing the mesoporous shells in an individual 3S-mesoUiO-66-NH₂ HoMS. **h** Scanning transmission electron microscopy (STEM) and elemental mapping images for Zr, C, and N elements. **i** XRD patterns, **j** N₂ sorption isotherms, and **k** pore size distributions of 3S-mesoUiO-66-NH₂ HoMSs and microporous UiO-66-NH₂ crystals.

precursors. TEM image reveals that the obtained MOF particles possess a core-shell architecture consisting of worm-like mesopores in the core and radially cylindrical mesochannels in the shell (Fig. 2d and Supplementary Fig. 11). We speculate that the two soft templates ODMB and F127 are sequentially assembled with the MOF precursor, resulting in dual-mesopore core-shell MOF particles with a less stable core and a more stable shell. The competitive interaction between ODMB and Zr-oxo clusters ($[Zr_6O_4(OH)_4]^{12+}$), as well as F127 and Zr-oxo clusters, respectively, influence the formation of two types of mesoporous MOFs. Besides, we further observe that the chemical etching of the mesoporous core is ascribed to the competition coordination between bidentate ligand BDC-NH₂ and monodentate ligand etching agents, including formic acid, acetic acid, and propanoic acid, to Zr-oxo clusters (Supplementary Fig. 12). The selective etching of mesostructured MOF cores could not occur in hydrochloric acid solution. At last, when using the dual-mesopore core-shell particles as seeds to

react with additional soft templates ODMB/F127 and MOF precursors twice, two additional layers of dual-mesopore UiO-66-NH₂ epitaxially grew (Fig. 2e and Supplementary Fig. 13). The newly formed MOF layers also possess a core-shell mesoporous structure similar to the dual-mesopore seed. When treated with acetic acid solution, the inner part of each layer with higher defect density is preferentially etched, while the outer part with higher crystallinity retains, leading to the formation of triple-shell hollow structures with highly accessible mesochannels (Figs. 1 and 2f).

On the basis of the above observations, we propose a scheme to illustrate the evolution pathway to hollow multi-shell mesoporous MOF particles (Fig. 2g). Compared with the electrostatic interaction among hydrophilic segments of Pluronic F127, Cl⁻ ions, and Zr-oxo clusters[42,43], the carboxylate functionality in ODMB molecules permits stronger coordination interaction with Zr-oxo clusters. Therefore, the interior part of mesostructured MOF with disordered worm-like pores

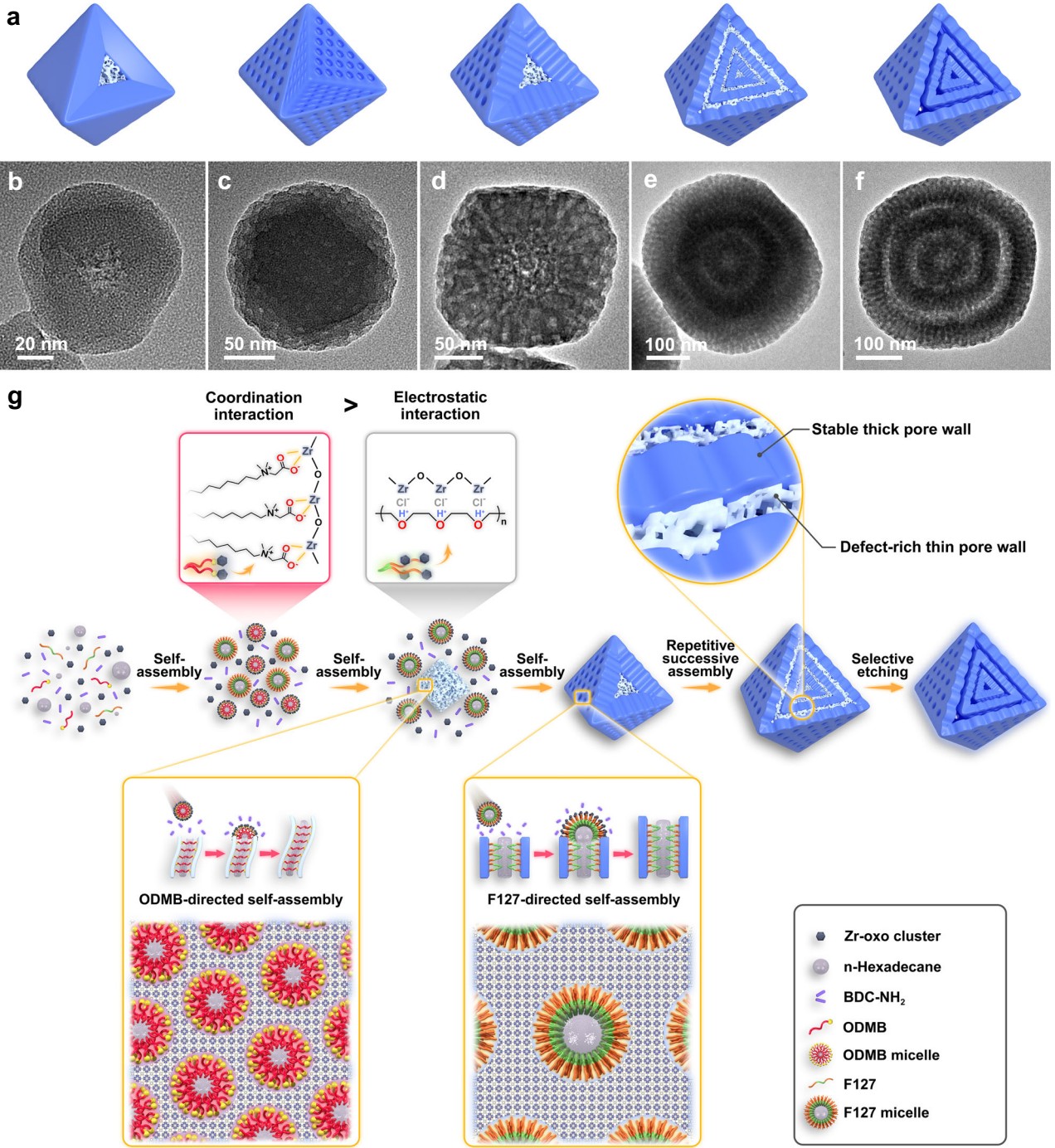

**Fig. 2 | Formation mechanism of 3S-mesoUiO-66-NH₂ HoMSs through dual-template-directed successive assembly and selective etching. a** Schematic models and their corresponding TEM images of UiO-66-NH₂ nanostructures prepared by (**b**) ionic surfactant ODMB-directed, **c** polymer surfactant F127-directed, **d** ODMB/F127 dual-template-directed successive assembly, **e** dual-template-directed successive assembly three times, and (**f**) chemical etching of the multilayer MOF precursor particle in acetic acid solution. **g** Formation mechanistic details of the dual-template-directed successive assembly and selective chemical etching.

is formed preferentially by cooperative self-assembly of ODMB and MOF precursors at the beginning of the reaction (Supplementary Fig. 14a). The white domains, as indicated by yellow circles, are occupied by OMDB surfactants, while the black domains, marked by red circles, represent the MOF pore wall (Supplementary Fig. 14b). Fourier transform infrared spectroscopy (FTIR) analysis reveals the coordination interaction of the Zr-oxo clusters to the ligands and ODMB surfactants (Supplementary Fig. 14c). The band centered at 580 cm⁻¹ is assigned to the characteristic Zr-(OC) asymmetric stretching[44].

Besides, the strong absorptions at 2925 and 2850 cm⁻¹ are caused by the C-H stretching vibration of methyl and methylene groups of ODMB. Large amounts of crystalline defects may exist in disordered thin mesopore walls with low crystallinity (Supplementary Fig. 14d). As the reaction time proceeds, ODMB templates gradually exhaust, and then F127-directed self-assembly starts by the epitaxial growth of mesoporous MOF layer with radically oriented mesochannels on the initially formed MOF cores (Supplementary Fig. 15a). The highly open mesochannels with thick walls and high crystallinity can be discerned

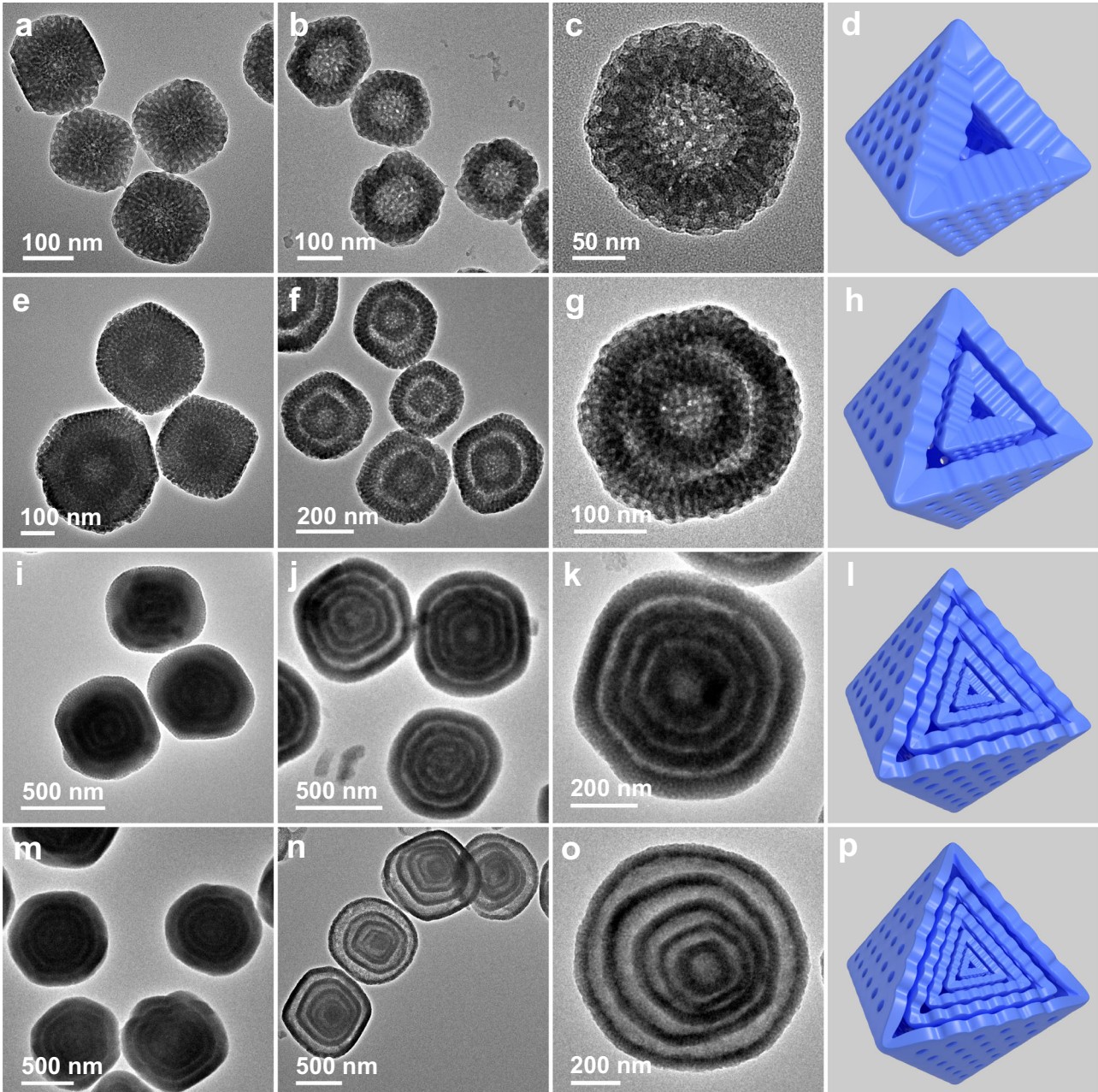

**Fig. 3 | Growth time of inhomogeneous MOF layers controlling the number of mesoporous shells.** TEM images of MOF precursor particles with (**a**) single, (**e**) double, (**i**) quadruple, and (**m**) quintuple inhomogeneous layers. TEM images of the corresponding products after etching treatment and schematic models: **b–d** 1S-mesoUiO-66-NH$_2$, **f–h** 2S-mesoUiO-66-NH$_2$, **j–l** 4S-mesoUiO-66-NH$_2$, and **n–p** 5S-mesoUiO-66-NH$_2$ HoMSs.

clearly (Supplementary Fig. 15b). Inhomogeneous mesoporous UiO-66-NH$_2$ particles have been successfully synthesized through ODMB/F127-directed sequence assembly. By continuously increasing the amount of ODMB from 0.05 g, to 0.1 g, then to 0.2 g, while keeping the amount of F127 constant, the diameter ratio of the interior core with disordered worm-like pores to the whole MOF particle varies accordingly, from 0.41, to 0.55, and finally to 0.73, further confirming ODMB/F127-directed sequence assembly mechanism (Supplementary Fig. 16). Afterwards, MOF precursor particles with triple inhomogeneous layers of alternating two mesostructured MOFs can be produced by repeating the growth process three times. At last, the defect-rich layers can be preferentially etched by acetic acid to produce hollow triple-shell mesoporous MOF particles.

Based on the growth mechanism mentioned above, the number of mesoporous shells is equal to the growth time of inhomogeneous

mesoporous MOF layers. The diameter of the MOF precursor particles increases with continuous growth of dual-mesopore MOFs for 1, 2, 4, and 5 times (Fig. 3a, e, i, m and Supplementary Figs. 11, 17–19). These particles can be converted into hollow single-shell mesoporous particles (1S-mesoUiO-66-NH$_2$ HoMSs), hollow double-shell mesoporous particles (2S-mesoUiO-66-NH$_2$ HoMSs), hollow quadruple-shell mesoporous particles (4S-mesoUiO-66-NH$_2$ HoMSs), and hollow quintuple-shell mesoporous particles (5S-mesoUiO-66-NH$_2$ HoMSs) after the acetic acid treatment (Fig. 3b–d, f–h, j–l, n–p). The high-magnification TEM images show that all MOF particles possess radially oriented mesochannels. Compared with their precursor particles with alternating layers of dual-mesopore MOFs, each shell in mesoUiO-66-NH$_2$ HoMSs retains the original size and pore structure of the robust mesoporous layers formed by F127-directed self-assembly. The SEM images further reveal the uniform diameter and morphology of the

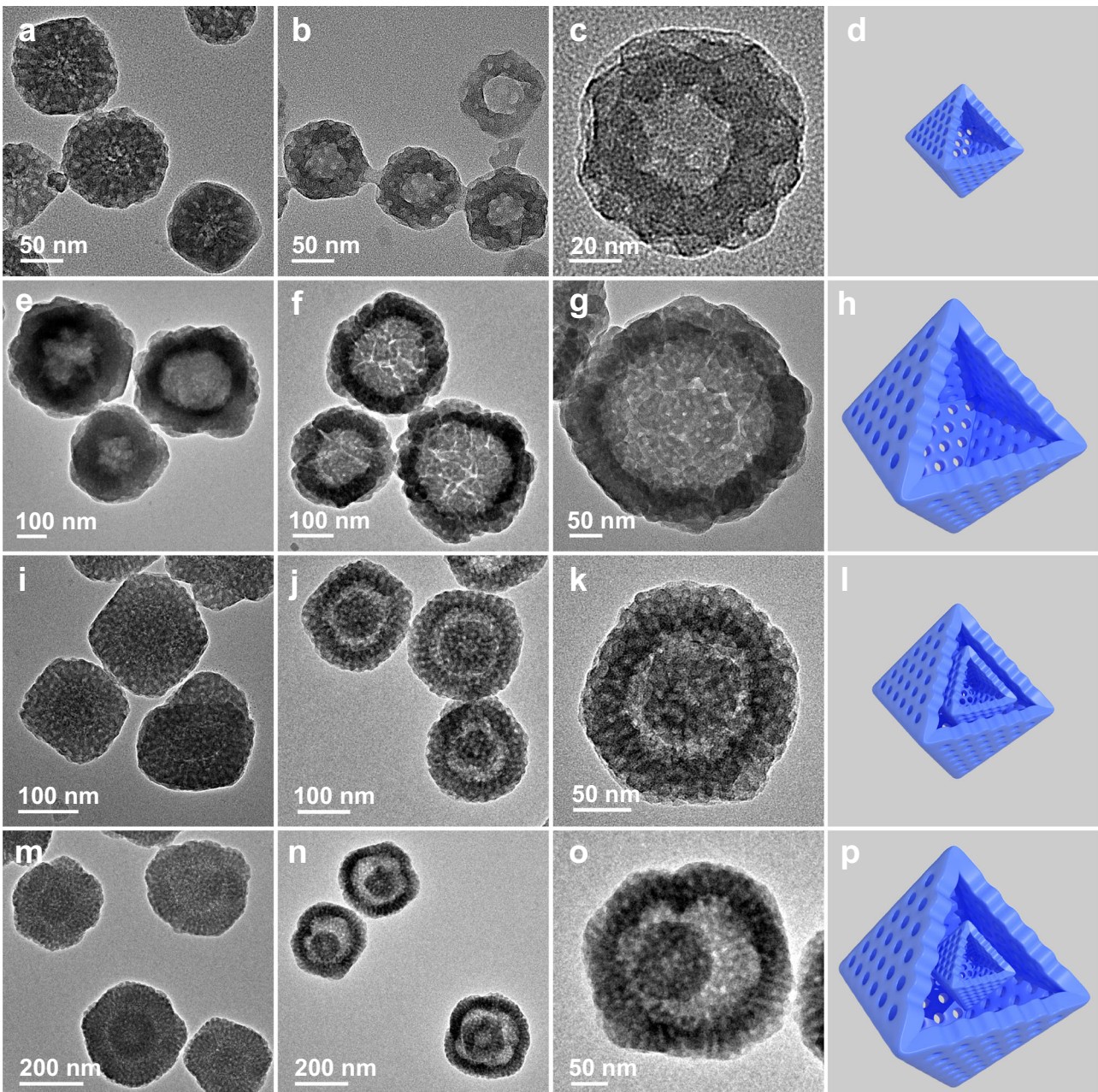

**Fig. 4 | Regulation of shell thickness, interior cavity size, and inter-shell space of mesoporous MOF particles.** TEM images of (**a**, **e**) single-layer and (**i**, **m**) double-layer MOF precursor particles with different thicknesses of inhomogeneous MOF layers, TEM images of the corresponding products after etching treatment and products (Supplementary Fig. 20). By further increasing the growth times of inhomogeneous mesoporous MOF layers, we were able to obtain hollow sextuple-shell mesoporous particles (6S-mesoUiO-66-NH$_2$ HoMSs) and septuple-shell mesoporous particles (7S-mesoUiO-66-NH$_2$ HoMSs) (Supplementary Fig. 21).

The shell thickness, interior cavity size, and inter-shell space of mesoporous UiO-66-NH$_2$ HoMSs can be precisely tailored at the nanoscale by tuning the nucleation kinetics of inhomogeneous dual-mesopore MOF precursors. During the formation of the single-layer MOF precursor particles, the nucleation kinetics is strongly associated with the concentration of acetic acid in the reaction system due to the competitive coordination of BDC-NH$_2$ and Zr-oxo clusters, as well as acetic acid and Zr-oxo clusters, respectively. By increasing the concentration of acetic acid from 1.94 M to 3.01 M, the diameter of the

schematic models: 1S-mesoUiO-66-NH$_2$ HoMSs prepared with the acetic acid concentrations of (**b**–**d**) 1.94 M and (**f**–**h**) 3.01 M, and 2S-mesoUiO-66-NH$_2$ HoMSs synthesized with (**j**–**l**) 80% and (**n**–**p**) 20% of single-layer MOF precursor particles prepared with the acetic acid concentration of 1.94 M in one pot as the cores.

single-layer MOF precursor particles increases from 90 to 360 nm (Fig. 4a, e and Supplementary Figs. 22,23). After etching, the shell thickness of the formed 1S-mesoUiO-66-NH$_2$ HoMSs can be varied from 30 to 60 nm, and the interior cavity size is tuned in the range from 30 to 240 nm (Fig. 4b–d, f–h). During the growth process of the second layer, the interfacial nucleation and growth kinetics of MOF layers is greatly influenced by the amount of single-layer MOF cores. When using 80% and 20% of single-layer MOF particles prepared with the acetic acid concentration of 1.94 M in one pot as the cores (Fig. 4a and Supplementary Fig. 22), the diameter of the double-layer MOF precursor particles prepared in 2.50 M acetic acid solution increases from 170 to 280 nm (Fig. 4i, m and Supplementary Figs. 24, 25). After etching, the thickness of the formed MOF layers increases from 40 to 55 nm (Fig. 4j–l, n–p). Accordingly, the space between the two shells of

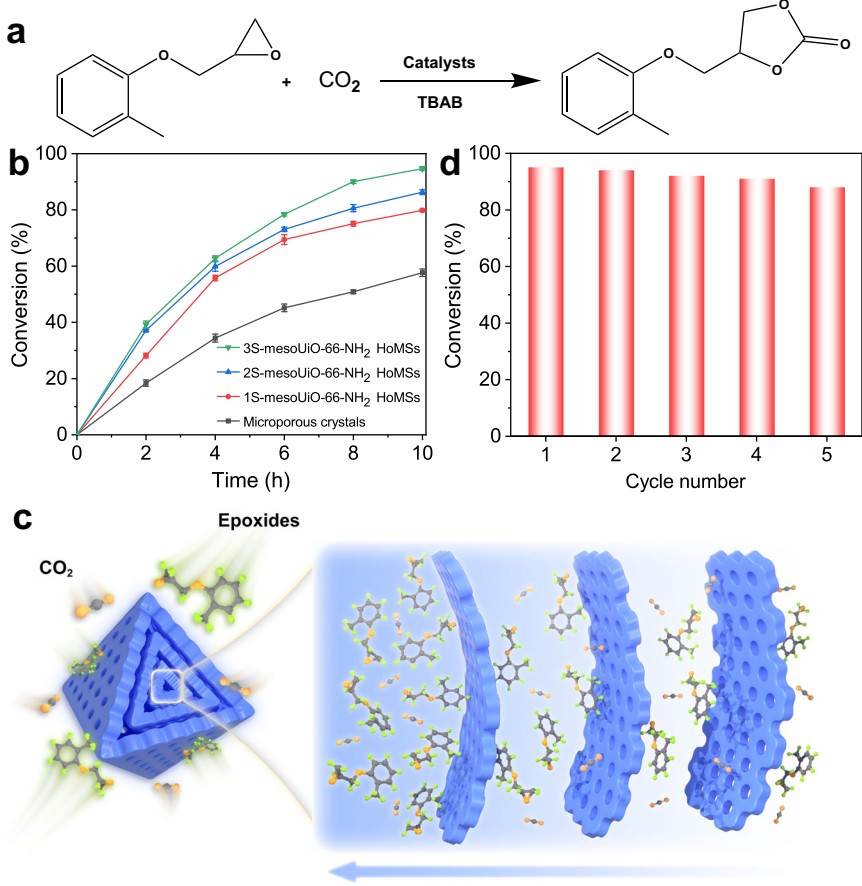

**Fig. 5 | CO$_2$ cycloaddition catalytic performance. a** Catalytic cycloaddition reactions between CO$_2$ and glycidyl-2-methylphenyl ether. **b** Kinetic profiles for CO$_2$ cycloaddition reaction by different catalysts over 10 h. Error bars represent standard deviation. **c** Schematic illustration of reactant enrichment in a 3S-mesoUiO-66-NH$_2$ HoMS. **d** The conversion of 5 recycle runs with 3S-mesoUiO-66-NH$_2$ HoMSs as the catalyst.

the resultant 2S-mesoUiO-66-NH$_2$ HoMSs varies from 25 to 75 nm after etching (Fig. 4j–l, n–p).

In addition, the space location of the microporous and mesoporous MOF shell can be well controlled by tuning the amount of F127 in the continuous assembly processes. When using F127 amounts of 0 g and 0.05 g in the two-step epitaxial growth of MOF layers, double-shell MOF particles consisting of inner microporous shells and outer mesoporous shells are obtained after chemical etching. On the contrary, when the F127 amounts of 0.05 g and 0 g are employed for the two-step epitaxial growth process, the resultant double-shell MOF particles with inner mesoporous shells and outer microporous shells are generated (Supplementary Fig. 26).

Moreover, we have verified the universality of methodology and growth mechanism by successfully synthesizing hollow multi-shell mesoporous Hf-UiO-66-NH$_2$ and MOF-801 particles. Following the similar growth mechanism mentioned above, inhomogeneous mesoporous Hf-UiO-66-NH$_2$ and MOF-801 precursor particles could also be formed by ODMB/F127-directed sequence assembly. By repeating such dual-template-directed assembly processes and subsequent selective etching, hollow double-shell mesoporous Hf-UiO-66-NH$_2$ and MOF-801 particles could be prepared (Supplementary Figs. 27–30).

To demonstrate the structural advantage of the mesoporous UiO-66-NH$_2$ HoMSs, we first investigate their accumulation capacities for the organic molecules. When using methylene blue as a model molecule, 3S-mesoUiO-66-NH$_2$ HoMSs exhibit faster adsorption kinetics and higher adsorption capacity than conventional microporous UiO-66-NH$_2$ crystals (Supplementary Fig. 31). Moreover, we observe that the adsorption rate and capacity increase along with the number of

layers in the mesoUiO-66-NH$_2$ HoMSs. The enhanced accumulation of the organic molecules may facilitate the catalytic kinetics. Besides, UiO-66-NH$_2$ framework features abundant Lewis acidic sites and amino functionalized ligands that facilitate CO$_2$ adsorption. Therefore, the CO$_2$ cycloaddition of epoxides to high-value cyclic carbonate products, was chosen as a probe reaction (Fig. 5a)[45]. Figure 5b shows the conversion for CO$_2$ cycloaddition reactions on conventional UiO-66-NH$_2$ microporous crystals, 1S-mesoUiO-66-NH$_2$, 2S-mesoUiO-66-NH$_2$, and 3S-mesoUiO-66-NH$_2$ HoMSs. The UiO-66-NH$_2$ hollow particles with open mesochannels in each shell exhibit higher activity than microporous MOF crystals. The catalytic activity increases along with the number of layers in the mesoUiO-66-NH$_2$ HoMSs. The conversion of glycidyl 2-methylphenyl ether can reach 95% in 10 h for the 3S-mesoUiO-66-NH$_2$ HoMSs. In contrast, the microporous UiO-66-NH$_2$ catalysts show low conversion rate of 58% in the same period. The discrepancy in catalytic activity among these MOF samples could be attributed to the presence of mesoporous structure and compartments between mesoporous layers, which may promote the accumulation of organic reactants and enhance catalytic kinetics (Fig. 5c)[30,46–49]. Moreover, recycling experiments are further performed with the 3S-mesoUiO-66-NH$_2$ HoMSs to investigate their catalytic stability. The catalyst shows no significant changes in terms of the catalytic activity after successive reuses of five cycles (Fig. 5d). In accordance with stability test, the mesoporous nanostructure and crystallinity of the reused catalyst remain unchanged after five reaction cycles compared with those of the fresh catalyst (Supplementary Fig. 32). To demonstrate the difference in the mass diffusion between 3S-mesoUiO-66-NH$_2$ HoMSs and microporous MOF crystals, we

further examine their performances in $CO_2$ cycloaddition reactions with two different epoxides propylene oxide and 1,2-epoxydodecane substituted with methyl and decyl groups, respectively. As expected, the 3S-mesoUiO-66-$NH_2$ catalyst shows higher conversion rate than microporous MOF catalyst (Supplementary Fig. 33). Distinct from the catalytic processes involving smaller substrate propylene oxide, the 3S-mesoUiO-66-$NH_2$ catalyst exhibits an almost twofold increase in the conversion of 1,2-epoxydodecane with larger substituted functional groups compared to the microporous MOF catalyst. This result suggests that mesoporous multi-shell nanostructures could enhance the mass transfer of larger molecules in the catalytic reactions, thereby improving their catalytic efficiency. The catalytic products of all epoxy compounds by $CO_2$ cycloaddition are determined by [1]H nuclear magnetic resonance ([1]H NMR) spectroscopy, and the conversion rates are calculated based on the peak area (Supplementary Figs. 34–36). The high catalytic performance of the 3S-mesoUiO-66-$NH_2$ HoMSs is ascribed to high accumulation capability of reactants and abundant accessible Lewis acidic sites in the open compartments of the multi-shell highly stable Zr-based MOF nanoarchitecture. A summary and comparison of the performances of different MOF catalysts in $CO_2$ cycloaddition reactions have been presented in Supplementary Table 1, where the 3S-mesoUiO-66-$NH_2$ HoMSs exhibit commendable catalytic performance as evidenced by the findings of this study. Furthermore, the mesoUiO-66-$NH_2$ HoMSs with highly open mesochannels are ideal substrates for the immobilization of metal nanoparticles. As an illustration, when Au nanoparticles are introduced into 3S-mesoUiO-66-$NH_2$ HoMSs, they exhibit excellent activity and recycling stability in hydrogenation reaction (Supplementary Fig. 37).

In summary, a dual-template-directed successive assembly approach has been developed to synthesize mesoporous UiO-66-$NH_2$ HoMSs with highly accessible mesochannels. The key feature of this strategy lies in the rational construction of MOF precursor particles with multiple alternating layers of two different mesostructured MOFs via continuous dual-template-directed assembly and their difference in framework stability upon chemical etching. The shell number of the mesoporous UiO-66-$NH_2$ HoMSs can be precisely controlled from 1 to 7 by adjusting the growth times of dual-mesopore MOFs. The shell thickness and cavity sizes of the multi-shelled mesoporous MOF particles can be facilely modulated. Moreover, the multi-shell inorganic-organic hybrid framework possesses radially oriented mesochannels (7.6 nm), and large pore volume (0.64 $cm^3 g^{-1}$). Benefiting from open compartments, abundant accessible active sites, and unique multi-shelled hollow structure, the mesoporous UiO-66-$NH_2$ HoMSs manifest enhanced catalytic activity and good stability in the $CO_2$ cycloaddition reaction. The present methodology may significantly expand the toolbox for rational construction of mesoporous and topologically complex MOF nanostructures for a wide range of applications.

# Methods
## Chemicals
Zirconyl chloride octahydrate ($ZrOCl_2 \cdot 8H_2O$, Aladdin), 2-aminoterephthalic acid (BDC-$NH_2$, 98%, Aladdin), F127 ($M_w$ = 12600, $EO_{106}PO_{70}EO_{106}$, Sigma-Aldrich), acetic acid (Sinopharm Chemical Reagent Company), formic acid (Aladdin), octadecyl dimethyl betaine (ODMB, 30% active matter, Chengdu Ruisi Reagent), n-hexadecane, (98%, Beijing InnoChem Science & Technology Co., Ltd.), glycerol (Sinopharm Chemical Reagent Company), glycidyl-2-methylphenyl ether (Aladdin), zirconium (IV) chloride ($ZrCl_4$, Aladdin), N,N-dimethylformamide (DMF, Sinopharm Chemical Reagent Company) tetrabutylammonium bromides (TBAB, 99%, Aladdin), methylene blue (MB, 98.5%,Tianjin Tianli Reagent), Hafnium(IV) chloride ($HfCl_4$, 99.5%, Macklin) fumaric acid (99.5%, Aladdin), deionized (DI) water from Milli-Q integral water purification system (Millipore, 18.2 $M\Omega \cdot cm^{-1}$).

## Synthesis of UiO-66-$NH_2$ dual-mesopore core-shell particles
Typically, 50 mg of ODMB and 50 mg of F127 were dissolved in 6.0 mL of deionized water, and then 1.0 mL of acetic acid was added. The mixture was stirred for 15 min to form a homogeneous solution. Subsequently, 161 mg of $ZrOCl_2 \cdot 8H_2O$, 0.15 mL of n-hexadecane, and 50 mg of BDC-$NH_2$ were added into above mixture. Then, the mixture was stirred at 45 °C for 10 h. The products were collected through centrifugation and then washed with DMF and ethanol. Finally, the products were redispersed in 2.0 mL of $H_2O$ for further use (suspension A).

## Synthesis of hollow single-shell mesoporous UiO-66-$NH_2$ particles (1S-mesoUiO-66-$NH_2$ HoMSs)
In a typical procedure, 1.0 mL of suspension A was well dispersed in a solution composed of 6.0 mL of $H_2O$ and 3.0 mL of acetic acid by magnetic stirring. After 6 h of stirring at 25 °C, the products were collected by centrifugation and washed several times in DMF and ethanol. To remove the template, the as-synthesized sample was soaked in ethanol for two days at 50 °C, during which time the ethanol was changed every day. After dried at 80 °C for 12 h, the 1S-mesoUiO-66-$NH_2$ were obtained.

## Synthesis of hollow double-shell mesoporous UiO-66-$NH_2$ particles (2S-mesoUiO-66-$NH_2$ HoMSs) and hollow triple-shell mesoporous UiO-66-$NH_2$ particles (3S-mesoUiO-66-$NH_2$ HoMSs)
Briefly, 50 mg of ODMB, 50 mg of F127, 0.15 g of glycerol, and 1.0 mL of acetic acid were dissolved in 5.0 mL of deionized water. Afterwards, 110 mg of $ZrOCl_2 \cdot 8H_2O$, 1.0 mL of suspension A as seeds, 0.15 mL of n-hexadecane, and 50 mg of BDC-$NH_2$ were added to the reaction system. The obtained mixture was continuously stirred at 50 °C for 8 h. The product MOFs with double inhomogeneous layers was separated by centrifugation, washed with DMF and ethanol several times and then redispersed in 2 mL of $H_2O$ for further use (suspension B). The MOF precursor particles with triple inhomogeneous layers was prepared by a procedure similar to that used for the MOFs with double inhomogeneous layers, except that suspension B was used as seeding. The 2S-mesoUiO-66-$NH_2$ and 3S-mesoUiO-66-$NH_2$ were synthesized by chemical etching of corresponding MOF precursor particles with acetic acid.

## Synthesis of microporous UiO-66-$NH_2$ crystals
Typically, 55.92 mg of $ZrCl_4$, 43.44 mg of BDC-$NH_2$, 6 mL of acetic acid were dissolved in 30 mL of DMF. The mixture was then sealed up and heated in an oven at 120 °C for 24 h. The as-synthesized UiO-66-$NH_2$ crystals were washed with DMF and ethanol, and then dried in an oven at 80 °C for 12 h.

## Characterization
The powder X-ray diffraction measurements were performed on a Rigaku D-Max 2550 diffractometer by using Cu Kα radiation. Scanning electron microscopy images were measured with JEOL JSM-7800F. The transmission electron microscopy images and the elemental mapping were measured with a Tecnai F20 electron microscope. Nitrogen adsorption/desorption measurements were carried out on a Micromeritics ASAP 3-flex analyzer at 77 K after the samples were degassed at 100 °C under vacuum. Fourier-transform infrared (FTIR) spectra of the samples dispersed in KBr pellets were measured on a PerkinElmer spectrum 430 FT-IR spectrometer. [1]H nuclear magnetic resonance spectroscopy (NMR) was recorded at room temperature using a Bruker AVANCE NEO spectrometer operating at frequencies of 400 MHz for [1]H.

## Dye adsorption studies
Dye adsorption efficiency of UiO-66-$NH_2$ were evaluated in 100 mg/L solutions of MB. The adsorbate uptake $q_t$ (mg/g) into the nanoparticles

at time t (min) were measured using:

$$q_t = \frac{(C_0 - C_t)V}{m} \qquad (1)$$

where $C_0$ and $C_t$ (mg/L) are dye concentrations at the start and any time t (min), m (g) is adsorbent mass, and V (L) represents the dye solution volume.

## Catalytic reaction

In a typical reaction, an autoclave was charged with 55 mg of catalysts, 0.98 g of glycidyl-2-methylphenyl ether, 50 mg of tetrabutylammonium bromide (TBAB) and 3 mL of DMF. After sealing, the assembled autoclave was purged of air three times with $CO_2$. After the $CO_2$ (0.5 MPa) was introduced, the reaction mixture was stirred at 65 °C for 10 h. After the reaction, unreacted $CO_2$ was slowly released and the catalyst was separated by centrifugation, and the supernatant was analyzed by $^1H$ NMR to identify the reaction products.

## Data availability

All the data generated in this study are provided in the main text and Supplementary Information. Source data are provided with this paper.

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

## Acknowledgements

We thank the National Natural Science Foundation of China (Grant 22288101, 21920102005, and 21835002), the National Key Research and Development Program of China (Grant 2021YFA1501202, 2022YFA1503600), and the 111 Project (B17020) for supporting this work. Y.Y. acknowledges the financial support from the JST ERATO Yamauchi Materials Space-Tectonics Project and the Queensland node of the NCRIS-enabled Australian National Fabrication Facility (ANFF).

## Author contributions

H.X. and B.G. conceived the project. H.X. prepared materials, performed measurements and analysed the data. J.H. and B.Z. performed SEM characterizations. R.S. and G.C. performed TEM characterizations. G.Z. helped with some of the experiments and characterization. All authors discussed the results and commented on the manuscript. H.X. wrote the draft, Y.Y. and B.G. revised and finalized the manuscript. B.G. supervised the work.

## Competing interests

The authors declare no competing interests.
