## [Peer Review File · Nature Communications]

REVIEWER COMMENTS

Reviewer #1 (Remarks to the Author):

Xu et al. introduced a straightforward method for fabricating monodisperse multi-shell mesoporous MOF (UiO-66-NH₂) particles using a dual-template-directed successive assembly approach. This approach involves the selective etching of successively grown multi-layer MOFs that consist of alternating two types of mesostructured layers. By employing this strategy, it becomes possible to create multi-shell mesoporous UiO-66-NH₂ nanostructures with precise control over the number of shells, accessible mesochannels, generous pore volume, and adjustable particle diameters and chamber sizes. The synthesis method presented in the study exhibits considerable appeal. Thus, I suggest its acceptance after some revisions.

1. A Table comparing the catalytic performance of the TSMPs with state-of-art catalysts in literature is suggested to be added in SI.
2. The time-conversion curves of catalytic cycloaddition with the error bars should be provided.
3. The resolution of some pictures is too low, e.g. Figure 1i, j, k, Figure 5b,d, please update them.
4. How do the authors confirm the products? GC-MS data or NMR spectroscopy of the products are suggested to be given.
5. More evidence is suggested to be added to further prove the diffusion advantages of the mesoporous structure. For example, Thiele modulus Φ and effectiveness factor η on different structures. The following related references published recently on mesoporous structures would be helpful: ACS Catal. 2020, 10, 11, 5973–5978; Nat Commun 13, 2900 (2022).

Reviewer #2 (Remarks to the Author):

This manuscript reported a facile dual-template-directed successive assembly strategy to synthesize UiO-66-NH₂ multi-shell mesoporous particles. The utilisation of dual-template are very distinctive, and the proposed mechanism in the manuscript is interesting. However, there are some issue to be address by authours. Firstly, the insufficient innovation in structural characteristics and synthesis methods. What is the different between this manuscript and some reported word in the design of the multi-shelled structure. F127 is the common template in synthesis of mesoporous material. The strategy here is

generates from the fusion of the two strategies without distinctive conception. Secondly, the strategy is lack universality and extensibility in more types of MOFs. Lastly, the selection of catalytic reaction is too ordinary to highlight the advantages of the structure. Therefore, at current stage, these findings are not sufficient to support its publication in Nature Communications. In addition, if the following questions are answered, it will help readers better understand this work.

1. The synthesis method of multi-shell mesoporous MOFs in the manuscript is at a relatively low temperature. Further clarification is needed on whether the crystallinity, defects, and stability of MOF crystals will be affected.
2. Although the effectiveness of individual templates has been studied, the proposed mechanism of the dual templates need further verification with more experiment or characterization .
3. The catalysis analysis is incomplete. For example, what are the catalytic active centers? Does the acidic defect site in MOFs have catalytic activity? Does etching affect the catalytic performance of MOF structures?
4. For MOFs as catalyst, most of reports can use the uniform micropore of MOFs for high selectivity. However, in this manuscript, author should do some catalytic reaction to demonstrate that the mesopore MOF shell enhance the molecular diffusion as well as micropore can achieve the selectivity.
5. The strategy is lack universality and extensibility in more types of MOFs. you can explore more kinds of MOFs, such as HKUST, MIL, which can be used this strategy to structure multi-shell mesoporous MOFs.
6. What is the maximum number of layers that can be achieved using this strategy, seven layers? Nine layers?
7. Can you design the space location of the micropore shell and mesopore shell by this strategy?
8. There are many errors in the manuscript. For instance, the metal-organic framework should be corrected as metal–organic framework. The CO₂ and TiO₂ should be corrected as CO₂ and TiO₂ in the references.

Reviewer #3 (Remarks to the Author):

This work developed an interesting synthetic strategy for multi-shell mesoporous UiO-66-NH₂. The method can precisely control shell numbers and sizes of the multi-shell mesoporous MOFs, with the demonstration of structural enhancement of adsorption capacities and catalytic performance. This work will expand the toolbox for multi-shell MOFs construction and will be of general interest to the scientific community. Therefore, it is suitable for publication in Nature Communications after addressing several points.

1. In Figure 1a and Figure 3, the acetic acid etching treatment is the last step to form the multi-shells. Also in line 115-121, the authors describe the last step is selective etching in the formation process of UiO-66-NH₂ MSMPs. However, there are no etching procedure in Method sections. Please correct the confusing parts.
2. I am curious about the generality of the synthetic strategy. Could the method apply to other UiO-66? Could the method apply to other MOFs?
3. If the coordination interaction of ODMB is the main reason to form disordered worm-like pores in UiO-66-NH₂, is it possible to use long chain fatty acid to replace ODMB?
4. Can the authors provide evidences that the catalytic active sites in UiO-66-NH₂ are Lewis acid sites?
5. Even though some mesoporous structures show similar phenomena, Can the authors explain why the mesoporous layers can facilitate the accumulation of organic molecules?

Reviewer #4 (Remarks to the Author):

In this manuscript, Xu et al. developed a dual-template-directed assembly approach for the preparation of monodisperse hollow multi-shelled structures (HoMS) UiO-66-NH₂. After multiple growth process, the formation of multi-shells with controllable shell numbers and tunable particle diameters could be achieved by chemical etching. The highly accessible Lewis acidic sites and favored mass transfer within the multi-shelled nanostructures, triple-shelled UiO-66-NH₂ particles show the highest activity in CO₂ cycloaddition reaction. This work provide some new attempts in the synthesis of HoMS materials. There are still ambiguous points that should be further clarified, the minor revision is needed.

1. The multi-shell mesoporous UiO-66-NH₂ particles reported in this work featuring more than two individual shells with isolated internal cavities have been well defined and widely used as hollow multi-shelled structures (HoMS) in previous reports (Adv. Mater. 2019, 31, 1802874, Nat. Chem. Rev. 2020, 4, 159-168, and Angew. Chem. Int. Ed. 2023, e202302621). It is strongly recommended that the author revise the nomenclature of the materials to better describe their characteristics and maintain consistency with the terminology used in the field.
2. In the introduction, the authors claim "The UiO-66-NH₂ MSMPs process controllable shell numbers (1-4), tunable particle diameters (90-600 nm)..." It seems that only single-shell structure could be observed in 90 nm sized UiO-66-NH₂ samples. In other words, what is the minimum material size that can form a quadruple-shelled UiO-66-NH₂ structure?
3. According to the synthesis mechanism now proposed, how is the regulation of shell spacing and shell thickness realized?

4. As the author proposes this as a synthetic strategy, it is important to determine whether this strategy is universal. Can other MOF monomers be used to verify its effectiveness? What is the key prerequisite for realizing the dual-template-directed assembly?
5. Since several sized pores exist in the as-synthesized material, the roles of the different sized pores should be clarified.
6. Related to question 4, can the enhancement mechanism of mass transfer process be verified with substrates containing different functional groups or substrates with different sizes?
7. What is the reason for the lack of catalytic performance data for quadruple-shelled UiO-66-NH₂ samples?

Response to the reviewers' comments

Reviewer #1:

Xu et al. introduced a straightforward method for fabricating monodisperse multi-shell mesoporous MOF (UiO-66-NH₂) particles using a dual-template-directed successive assembly approach. This approach involves the selective etching of successively grown multi-layer MOFs that consist of alternating two types of mesostructured layers. By employing this strategy, it becomes possible to create multi-shell mesoporous UiO-66-NH₂ nanostructures with precise control over the number of shells, accessible mesochannels, generous pore volume, and adjustable particle diameters and chamber sizes. The synthesis method presented in the study exhibits considerable appeal. Thus, I suggest its acceptance after some revisions.

Comment 1: A Table comparing the catalytic performance of the TSMPs with state-of-art catalysts in literature is suggested to be added in SI.

Response: We are greatly thankful to the reviewer for the encouragement on our work and the valuable comments. We have compiled a table summarizing and comparing the performance of various MOF catalysts in CO₂ cycloaddition reactions (Supplementary Table 1). The corresponding discussion has been added on Page 11, highlighted in yellow.

Supplementary Table 1. Comparison of 3S-mesoUiO-66-NH₂ HoMSs with other reported MOF catalysts for the CO₂ cycloaddition reactions.

Catalyst	Temperature (°C)	Pressure (bar)	Time (h)	Conversion (%)	Reference
Gea-MOF-1	120	2	6	85	Nat. Chem. 2014 , 6, 673-680
FJI-H14	80	1	24	86	Nat. Commun. 2017 , 8, 1233
HKUST-1	25	1	48	65	J. Am. Chem. Soc. 2016 , 138, 2142-2145
(I-)Meim-UiO-66	120	1	24	33	Chem. Sci. 2017 , 8, 1570-1575
PCN-222	50	4	24	66	ChemCatChem 2018 , 10, 3506-3512
Co-MOF-74	100	2	4	96	Catal. Today 2009 , 148, 221-231

I ₂ HCP-5b	120	30	4	87	ChemSusChem 2020 , 13 , 341-350
MOF-892	25	1	60	70	ACS Appl. Mater. Interfaces 2018 , 10 , 733-744
In-MIL-68-NH ₂	150	0.8	8	74	ChemCatChem 2012 , 4 , 1725-1728
ZIF-67	100	10	15	92	New J. Chem. 2016 , 40 , 5170-5176
3S-mesoUiO-66-NH ₂	65	5	10	95	This work

Comment 2: The time-conversion curves of catalytic cycloaddition with the error bars should be provided.

Response: Per the suggestion of the reviewer, we have provided the time-conversion curves of the catalytic reactions with error bars. The corresponding discussion has been added on Page 10, highlighted in yellow.

Figure 5 | CO₂ cycloaddition catalytic performance. (a) Catalytic cycloaddition reactions between CO₂ and glycidyl-2-methylphenyl ether. (b) Kinetic profiles for CO₂ cycloaddition reactions by different catalysts over 10 h. (c) Schematic illustration of reactant enrichment in a 3S-mesoUiO-66-NH₂ HoMS. (d) The conversion of 5 recycle runs with 3S-mesoUiO-66-NH₂ HoMSs as the catalyst.

Comment 3: The resolution of some pictures is too low, e.g. Figure 1i, j, k, Figure 5b,d, please update them.

Response: Per the suggestion of the reviewer, we have updated the related panels in Figure 1 and Figure 5.

Figure 1 | Synthesis and characterization of the **3S-mesoUiO-66-NH₂ HoMSs**. (a) Schematic illustration of the synthesis process. (b-d) SEM and (e,f) TEM images of **3S-mesoUiO-66-NH₂ HoMSs**. (g) Magnified TEM image showing the mesoporous shells in an individual **3S-mesoUiO-66-NH₂ HoMSs**. (h) Scanning transmission electron microscopy (STEM) and elemental mapping images for Zr, C, and N elements. (i) XRD patterns, (j) N₂ sorption isotherms, and (k) pore size distributions of **3S-mesoUiO-66-NH₂ HoMSs** and microporous UiO-66-NH₂ crystals.

Figure 5 | CO₂ cycloaddition catalytic performance. (a) Catalytic cycloaddition reactions between CO₂ and glycidyl-2-methylphenyl ether. (b) Kinetic profiles for CO₂ cycloaddition reactions by different catalysts over 10 h. (c) Schematic illustration of reactant enrichment in a **3S-**

mesoUiO-66-NH₂ HoMS. (d) The conversion of 5 recycle runs with 3S-mesoUiO-66-NH₂ HoMSs as the catalyst.

Comment 4: How do the authors confirm the products? GC-MS data or NMR spectroscopy of the products are suggested to be given.

Response: Thank you for your valuable comment. The catalytic products of all epoxy compounds by CO₂ cycloaddition are determined by ¹H nuclear magnetic resonance (¹H NMR) spectroscopy, and the conversion rates are calculated based on the peak area (Supplementary Figs. 34-36). The corresponding discussion has been added on Page 10 and 11, highlighted in yellow.

Supplementary Fig. 34 ¹H NMR spectra (DMSO-d₆) of glycidyl-2-methylphenyl ether and the corresponding product under pure CO₂ atmosphere catalyzed by (a) 3S-mesoUiO-66-NH₂ HoMSs, (b) 2S-mesoUiO-66-NH₂ HoMSs, (c) 1S-mesoUiO-66-NH₂ HoMSs, and (d) microporous crystals.

Supplementary Fig. 35 ¹H NMR spectra (DMSO-d₆) of propylene oxide and the corresponding product under pure CO₂ atmosphere catalyzed by (a) 3S-mesoUiO-66-NH₂ HoMSs and (b) microporous crystals.

Supplementary Fig. 36 ^1H NMR spectra (DMSO- d_6) of 1,2-epoxydodecane and the corresponding product under pure CO_2 atmosphere catalyzed by (a) 3S-mesoUiO-66- NH_2 HoMSs and (b) microporous crystals.

Comment 5: More evidence is suggested to be added to further prove the diffusion advantages of the mesoporous structure. For example, Thiele modulus Φ and effectiveness factor η on different structures. The following related references published recently on mesoporous structures would be helpful: ACS Catal. 2020, 10, 11, 5973–5978; Nat Commun 13, 2900 (2022).

Response: Thanks very much for the valuable comment. To compare the differences in the mass diffusion behavior between 3S-mesoUiO-66- NH_2 HoMSs and microporous MOF crystals, we performed a gravimetric analysis on the diffusion of TMB under inert conditions. As expected, the uptake of TMB in the 3S-mesoUiO-66- NH_2 HoMSs is much faster than that in microporous crystals. We have also cited the two related studies you provided as Ref. 43 and 44.

TMB uptake curves of 3S-mesoUiO-66- NH_2 HoMSs and microporous MOF crystals measured by gravimetric analysis.

Reviewer #2:

This manuscript reported a facile dual-template-directed successive assembly strategy to synthesize UiO-66- NH_2 multi-shell mesoporous particles. The utilisation of dual-template are very distinctive, and the proposed mechanism in the manuscript is interesting. However, there are some issue to be address by authours. Firstly, the insufficient innovation in structural characteristics and synthesis methods. What is the different between this manuscript and some reported word in the design of the multi-shelled structure. F127 is the common template in synthesis of mesoporous material. The strategy here is generates from the fusion of the two strategies without distinctive conception. Secondly, the strategy is lacking universality and extensibility in more types of MOFs. Lastly, the selection of catalytic reaction is too ordinary to highlight the advantages of the structure. Therefore, at current stage, these findings are not sufficient to support its publication in Nature Communications. In addition, if the following questions are answered, it will help readers better understand this work.

Response: We are greatly thankful to the referee for the encouragement on our work and the excellent comments and questions. We hope the detailed discussion and relevant revision could address the issues raised by the reviewer and dispel the possible

hesitation and misunderstanding from the reviewer. Herein, we would like to restate the novelty and importance of our work as follows:

The multi-shell MOF particles have attracted special research interest due to their unique compartmentation environments, plentiful heterogeneous interfaces, and remarkable diversity of crystalline architectures. Although there are several important studies on the synthesis of specific MOF multi-shell particles with intrinsic microporous shells (*Angew. Chem. Int. Ed.* **2017**, *56*, 5512; *Angew. Chem. Int. Ed.* **2018**, *57*, 2110; *Nat Commun* **2017**, *8*, 14070), general method that can achieve various multi-shell MOF materials with mesoporous shells has not been reported yet. Furthermore, we present a novel dual-template-directed successive assembly mechanism, which markedly differs from the formation mechanism observed in previous reported multi-shell MOF particles or multi-shell mesoporous particles. Thanks to this unique growth mechanism, we are able to achieve multi-shell mesoporous MOF particles with controllable shell numbers, accessible mesochannels, tunable shell thickness and chamber sizes. To date, precise control over the structure of multi-shell mesoporous MOF particles has not been achieved yet.

Moreover, we have verified the universality of methodology and growth mechanism by successfully synthesizing hollow multi-shell mesoporous Hf-UiO-66-NH₂ and MOF-801 particles. Following the similar growth mechanism mentioned above, inhomogeneous mesoporous Hf-UiO-66-NH₂ and MOF-801 precursor particles could be formed by ODMB/F127-directed sequence assembly. By repeating such dual-template-directed assembly processes and subsequent selective etching, hollow double-shell mesoporous Hf-UiO-66-NH₂ and MOF-801 particles could be prepared (Supplementary Figs. 27-30). The corresponding discussion has been added on Page 9, highlighted in yellow.

Supplementary Fig. 27 TEM images of (a,b) dual-mesopore core-shell Hf-UiO-66-NH₂ precursor particles prepared with both ODMB and F127 templates, (c,d) single-shell mesoporous Hf-UiO-66-NH₂ particles, (e,f) double-inhomogeneous-layer Hf-UiO-66-NH₂ precursor particles, and (g,h) double-shell mesoporous Hf-UiO-66-NH₂ particles.

Supplementary Fig. 28 XRD pattern of double-shell mesoporous Hf-UiO-66-NH₂ particles.

Supplementary Fig. 29 TEM images of (a,b) dual-mesopore core-shell MOF-801 precursor particles prepared with both ODMB and F127 templates, (c,d) single-shell mesoporous MOF-801 particles, (e,f) double-inhomogeneous-layer MOF-801 precursor particles, and (g,h) double-shell mesoporous MOF-801 particles.

Supplementary Fig. 30 XRD pattern of double-shell mesoporous MOF-801 particles.

Lastly, catalyzed CO₂ cycloaddition with epoxides to produce carbonates is of great significance due to the wide applications of carbonates in pharmaceutical and electrochemical industries. To showcase the superiority of 3S-mesoUiO-66-NH₂ HoMSs in terms of mass diffusion compared to microporous MOF crystals, we evaluate their effectiveness in CO₂ cycloaddition reactions utilizing two different epoxides propylene oxide and 1,2-epoxydodecane substituted with methyl and decyl groups, respectively. As expected, the 3S-mesoUiO-66-NH₂ catalyst shows the higher conversion rates than microporous MOF catalyst (Supplementary Fig. 33). Distinct from the catalytic processes involving small substrate propylene oxide, the 3S-mesoUiO-66-NH₂ catalyst exhibits an almost twofold increase in the conversion rate of 1,2-epoxydodecane with larger substituted functional groups compared to the microporous MOF catalyst. This result suggests that mesoporous multi-shell nanostructures could enhance the mass transfer of larger molecules in the catalytic reactions, thereby improving their catalytic efficiency. The corresponding discussion has been added on Page 10, highlighted in yellow.

Supplementary Fig. 33 The conversion of propylene oxide and 1,2-epoxydodecane in the CO₂ cycloaddition reactions catalyzed by 3S-mesoUiO-66-NH₂ HoMSs and microporous MOF crystals.

Comment 1: The synthesis method of multi-shell mesoporous MOFs in the manuscript is at a relatively low temperature. Further clarification is needed on whether the crystallinity, defects, and stability of MOF crystals will be affected.

Response: Many thanks for the valuable comment. The sharp and intense diffraction peaks of 3S-mesoUiO-66-NH₂ HoMSs observed in the XRD pattern match well with the diffraction peaks of the microporous UiO-66-NH₂ crystals, suggesting that the crystallinity of multi-shell mesoporous MOFs is not affected in low temperature synthesis condition (Fig. 11). Moreover, compared with microporous UiO-66-NH₂ crystals, the micropore volume of the multi-shell mesoporous MOFs decreases from 0.32 to 0.25 cm³ g⁻¹ and the total volume increases from 0.44 to 0.64 cm³ g⁻¹, indicating that more defects exist in the 3S-mesoUiO-66-NH₂ HoMSs. Notably, thermogravimetric (TG) analysis curves show the influence of structural defects on the thermal stability of 3S-mesoUiO-66-NH₂ is negligible (Supplementary Fig. 3). The corresponding discussion has been added on Page 5, highlighted in yellow.

Supplementary Fig.3. TG curves of 3S-mesoUiO-66-NH₂ HoMSs and microporous UiO-66-NH₂ crystals.

Comment 2: Although the effectiveness of individual templates has been studied, the proposed mechanism of the dual templates need further verification with more experiment or characterization.

Response: Thank you very much for your valuable comment. We think the interior part of mesostructured MOF with disordered worm-like pores is formed preferentially by cooperative self-assembly of ODMB and MOF precursors at the beginning of the

reaction. As the reaction time proceeds, ODMB templates gradually exhaust, and then F127-directed self-assembly starts by the epitaxial growth of mesoporous MOF layer with radically oriented mesochannels on the initially formed MOF cores.

To further validate our speculation, we progressively increase the amount of ODMB from 0.05 g to 0.1 g, and then to 0.2 g, while keeping the amount of F127 constant. Consequently, the diameter ratio of the interior core with disordered worm-like pores to the whole MOF particle varies accordingly, changing from 0.41 to 0.55, and eventually reaching 0.73 (Supplementary Fig. 16). This observation provides confirmation of ODMB/F127-directed sequence assembly mechanism. The corresponding discussion has been added on Page 7, highlighted in yellow.

Supplementary Fig. 16 TEM and magnified TEM images of dual-mesopore core-shell UiO-66-NH₂ particles prepared with 0.05 g of F127 but different amount of ODMB: (a,b) 0.05 g, (c,d) 0.1 g, and (e,f) 0.2 g.

Comment 3: The catalysis analysis is incomplete. For example, what are the catalytic active centers? Does the acidic defect site in MOFs have catalytic activity? Does etching affect the catalytic performance of MOF structures?

Response: Many thanks for the comment. The Lewis acid metal centers in MOFs can promote the activation of the epoxide ring, while the functional groups in the ligands can act as basic sites improving the CO₂ affinity inside the pore (*Chem. Soc. Rev.* **2019**, *48*, 2783-2828, *Inorganics* **2021**, *9*, 81). A previous study reported the acidic defect site can provide additional catalytically active sites from undercoordinated zirconium nodes, leading to higher catalytic conversion in the cycloaddition reaction (*J. Mater. Chem. A* **2022**, *10*, 10051-10061).

To explore the effect of etching on the catalytic performance of MOF structures, we synthesized mesoporous MOF using F127 as the sole template (Fig. 2c) and evaluate the catalytic performance between etched and unetched samples. The conversion of glycidyl 2-methylphenyl ether can reach 74% in 10 h for the unetched samples and the etched samples show conversion of 77% in the same period. The results indicate that etching has a negligible effect on the catalytic performance.

Comment 4: For MOFs as catalyst, most of reports can use the uniform micropore of MOFs for high selectivity. However, in this manuscript, author should do some catalytic

reaction to demonstrate that the mesopore MOF shell enhance the molecular diffusion as well as micropore can achieve the selectivity.

Response: Many thanks for the comment. The inherent CO₂ absorbability, the exposed Lewis acid metal sites, and the confinement of the pore size make the MOF a promising heterogeneous catalyst for the cycloaddition of CO₂ with epoxides to form cyclic carbonates (*ACS Catal.* **2018**, *8*, 3194–3201; *Chem. Mater.* **2019**, *31*, 3, 1084–1091). MOF-catalyzed CO₂ cycloaddition of small substrates was carried out within the framework, while large ones cannot easily enter into the porous framework for catalytic reactions. Thus, the synthesized MOFs could exhibit size-dependent selectivity to different substrates on account of the confinement of the pore diameter (*J. Am. Chem. Soc.* **2016**, *138*, 2142–2145).

To demonstrate the enhanced molecular diffusion and the size-dependent selectivity of mesopore MOF shell, we further examine their performances in CO₂ cycloaddition reactions with two different epoxides propylene oxide and 1,2-epoxydodecane substituted with methyl and decyl groups, respectively. As expected, the 3S-mesoUiO-66-NH₂ catalyst shows the higher conversion rates than microporous MOF catalyst (Supplementary Fig. 33). Distinct from the catalytic processes involving smaller substrate propylene oxide, the 3S-mesoUiO-66-NH₂ catalyst exhibits an almost twofold increase in the conversion rate of 1,2-epoxydodecane with larger substituted functional groups compared to the microporous MOF catalyst. This result suggests that mesoporous multi-shell nanostructures can not only enhance the mass transfer of larger molecules but also maintain higher size-dependent selectivity toward smaller epoxides in catalytic CO₂ cycloaddition. The corresponding discussion has been added on Page 10, highlighted in yellow.

Supplementary Fig. 33 The conversion of propylene oxide and 1,2-epoxydodecane in the CO₂ cycloaddition reactions catalyzed by 3S-mesoUiO-66-NH₂ HoMSs and microporous MOF crystals.

Comment 5: The strategy is lack universality and extensibility in more types of MOFs. you can expore more kinds of MOFs, such as HKUST, MIL, which can be used this strategy to structure multi-shell mesoporous MOFs.

Response: Many thanks for the valuable comment. We have verified the universality of methodology and growth mechanism by successfully synthesizing hollow multi-shell mesoporous Hf-UiO-66-NH₂ and MOF-801 particles. Following the similar growth mechanism mentioned above, inhomogeneous mesoporous Hf-UiO-66-NH₂

and MOF-801 precursor particles could be formed by ODMB/F127-directed sequence assembly. By repeating such dual-template-directed assembly processes and subsequent selective etching, hollow double-shell mesoporous Hf-UiO-66-NH₂ and MOF-801 particles could be prepared (Supplementary Figs. 27-30). The corresponding discussion has been added on Page 9, highlighted in yellow.

Supplementary Fig. 27 TEM images of (a,b) dual-mesopore core-shell Hf-UiO-66-NH₂ precursor particles prepared with both ODMB and F127 templates, (c,d) single-shell mesoporous Hf-UiO-66-NH₂ particles, (e,f) double-inhomogeneous-layer Hf-UiO-66-NH₂ precursor particles, and (g,h) double-shell mesoporous Hf-UiO-66-NH₂ particles.

Supplementary Fig. 28 XRD pattern of double-shell mesoporous Hf-UiO-66-NH₂ particles.

Supplementary Fig. 29 TEM images of (a,b) dual-mesopore core-shell MOF-801 precursor particles prepared with both ODMB and F127 templates, (c,d) single-shell mesoporous MOF-801 particles, (e,f) double-inhomogeneous-layer MOF-801 precursor particles, and (g,h) double-shell mesoporous MOF-801 particles.

Supplementary Fig. 30 XRD pattern of double-shell mesoporous MOF-801 particles.

Comment 6: What is the maximum number of layers that can be achieved using this strategy, seven layers? Nine layers?

Response: Many thanks for the valuable comment. By further increasing the growth times of inhomogeneous mesoporous MOF layers, we were able to obtain quintuple-shell, sextuple-shell, and septuple-shell mesoporous particles (Fig. 3n-p and Supplementary Fig. 21). The method utilized in this study holds the potential to synthesize hollow multi-shell mesoporous MOFs with even more layers. The corresponding discussion has been added on Page 8, highlighted in yellow.

Figure 3 | Growth time of inhomogeneous MOF layers controlling the number of mesoporous shells. TEM images of MOF precursor particles with (a) single, (e) double, (i) quadruple, and (m) quintuple inhomogeneous layers. TEM images of the corresponding products after etching treatment and schematic models: (b-d) 1S-mesoUiO-66-NH₂, (f-h) 2S-mesoUiO-66-NH₂, (j-l) 4S-mesoUiO-66-NH₂, and (n-p) 5S-mesoUiO-66-NH₂ HoMSs.

Supplementary Fig. 21 TEM images of (a) 6S-mesoUiO-66-NH₂ and (b) 7S-mesoUiO-66-NH₂ HoMSs.

Comment 7: Can you design the space location of the micropore shell and mesopore shell by this strategy?

Response: Thanks very much for the valuable comment. The space location of the microporous and mesoporous MOF shell can be well controlled by tuning the amount of F127 in the continuous assembly processes. When using F127 amounts of 0 g and 0.05 g in the two-step epitaxial growth of MOF layers, double-shell MOF particles consisting of inner microporous shells and outer mesoporous shells are obtained after chemical etching. On the contrary, when the F127 amounts of 0.05 g and 0 g are employed for the two-step epitaxial growth process, the resultant double-shell MOF particles with inner mesoporous shells and outer microporous shells are generated (Supplementary Fig. 26). The corresponding discussion has been added on Page 9, highlighted in yellow.

Supplementary Fig. 26 TEM images and magnified TEM images of (a,b) double-shell UiO-66-NH₂ particles with inner microporous shells and outer mesoporous shells and (c,d) double-shell UiO-66-NH₂ particles with inner mesoporous layers and outer microporous layers.

Comment 8: There are many errors in the manuscript. For instance, the metal-organic

framework should be corrected as metal–organic framework. The CO₂ and TiO₂ should be corrected as CO₂ and TiO₂ in the references.

Response: Per the suggestion of the reviewer, we have revised the relevant parts in the manuscript accordingly.

Reviewer #3:

This work developed an interesting synthetic strategy for multi-shell mesoporous UiO-66-NH₂. The method can precisely control shell numbers and sizes of the multi-shell mesoporous MOFs, with the demonstration of structural enhancement of adsorption capacities and catalytic performance. This work will expand the toolbox for multi-shell MOFs construction and will be of general interest to the scientific community. Therefore, it is suitable for publication in Nature Communications after addressing several points.

Comment 1: In Figure 1a and Figure 3, the acetic acid etching treatment is the last step to form the multi-shells. Also in line 115-121, the authors describe the last step is selective etching in the formation process of UiO-66-NH₂ MSMPs. However, there are no etching procedure in Method sections. Please correct the confusing parts.

Response: We are greatly thankful to the reviewer for the encouragement on our work and the valuable comments. We have highlighted the etching procedure in the method sections of the manuscript in yellow on Page 17.

Comment 2: I am curious about the generality of the synthetic strategy. Could the method apply to other UiO-66? Could the method apply to other MOFs?

Response: Thank you very much for the valuable comment. We have verified the universality of methodology and growth mechanism by successfully synthesizing hollow multi-shell mesoporous Hf-UiO-66-NH₂ and MOF-801 particles. Following the similar growth mechanism mentioned above, inhomogeneous mesoporous Hf-UiO-66-NH₂ and MOF-801 precursor particles could be formed by ODMB/F127-directed sequence assembly. By repeating such dual-template-directed assembly processes and subsequent selective etching, hollow double-shell mesoporous Hf-UiO-66-NH₂ and MOF-801 particles could be prepared (Supplementary Figs. 27-30). The corresponding discussion has been added on Page 9, highlighted in yellow.

Supplementary Fig. 27 TEM images of (a,b) dual-mesopore core-shell Hf-UiO-66-NH₂ precursor particles prepared with both ODMB and F127 templates, (c,d) single-shell mesoporous Hf-UiO-66-

NH₂ particles, (e,f) double-inhomogeneous-layer Hf-UiO-66-NH₂ precursor particles, and (g,h) double-shell mesoporous Hf-UiO-66-NH₂ particles.

Supplementary Fig. 28 XRD pattern of double-shell mesoporous Hf-UiO-66-NH₂ particles.

Supplementary Fig. 29 TEM images of (a,b) dual-mesopore core-shell MOF-801 precursor particles prepared with both ODMB and F127 templates, (c,d) single-shell mesoporous MOF-801 particles, (e,f) double-inhomogeneous-layer MOF-801 precursor particles, and (g,h) double-shell mesoporous MOF-801 particles.

Supplementary Fig. 30 XRD pattern of double-shell mesoporous MOF-801 particles.

Comment 3: If the coordination interaction of ODMB is the main reason to form disordered worm-like pores in UiO-66-NH₂, is it possible to use long chain fatty acid to replace ODMB?

Response: Many thanks for the valuable comment. When replacing ODMB with octadecanoic acid, inhomogeneous mesoporous MOF core-shell nanostructure with the worm-like mesostructured core cannot be formed. The result could be attributed to the fact that long-chain fatty acids become insoluble in water under acidic condition, and cannot function as soft templates for the cooperative self-assembly to generate mesoporous MOFs.

TEM (a) and magnified TEM (b) images of UiO-66-NH₂ mesoporous particles prepared with octadecanoic acid and F127 templates.

Comment 4: Can the authors provide evidences that the catalytic active sites in UiO-66-NH₂ are Lewis acid sites?

Response: Many thanks for the valuable comment. MOFs have been developed as efficient catalysts for the synthesis of cyclic carbonates from CO₂ and epoxides (*J. Mater. Chem. A* **2022**, *10*, 10051-10061; *Green Chem.* **2016**, *18*, 4086-4091). A previous study reported the Lewis acid metal centers in MOFs can promote the activation of the epoxide ring, while the functional groups in the ligands can act as basic sites improving the CO₂ affinity inside the pore (*Chem. Soc. Rev.* **2019**, *48*, 2783-2828, *Inorganics* **2021**, *9*, 81).

Comment 5: Even though some mesoporous structures show similar phenomena, Can the authors explain why the mesoporous layers can facilitate the accumulation of organic molecules?

Response: Many thanks for the comment. The hollow structure creates a confined space, and outer reactants would continuously diffuse into the interior of the hollow structure directionally driven by the concentration gradient and/or capillary-like effect (*Natl. Sci. Rev.* **2023**, DOI: 10.1093/nsr/nwad201). This enrichment effect could hinder the diffusion of inner reactants out of the cavity and increase the local concentration of reactant around the active sites, resulting in an accelerated reaction rate and an improved catalytic performance (*ACS Catal.* **2018**, *8*, 1218–1226).

Reviewer #4:

In this manuscript, Xu et al. developed a dual-template-directed assembly approach for the preparation of monodisperse hollow multi-shelled structures (HoMS) UiO-66-NH₂. After multiple growth process, the formation of multi-shells with controllable shell numbers and tunable particle diameters could be achieved by chemical etching. The highly accessible Lewis acidic sites and favored mass transfer within the multi-shelled nanostructures, triple-shelled UiO-66-NH₂ particles show the highest activity in CO₂ cycloaddition reaction. This work provide some new attempts in the synthesis of HoMS materials. There are still ambiguous points that should be further clarified, the minor revision is needed.

Comment 1: The multi-shell mesoporous UiO-66-NH₂ particles reported in this work featuring more than two individual shells with isolated internal cavities have been well defined and widely used as hollow multi-shelled structures (HoMS) in previous reports (Adv. Mater. 2019, 31, 1802874, Nat. Chem. Rev. 2020, 4, 159-168, and Angew. Chem. Int. Ed. 2023, e202302621). It is strongly recommended that the author revise the nomenclature of the materials to better describe their characteristics and maintain consistency with the terminology used in the field.

Response: We are greatly thankful to the reviewer for the encouragement on our work and the valuable comments. We have modified the nomenclature of the mesoporous UiO-66-NH₂ hollow multi-shelled particles to *n*S-mesoUiO-66-NH₂ HoMSs, where *n* indicates the number of shells. The corresponding discussion has been added on Page 4, highlighted in yellow.

Comment 2: In the introduction, the authors claim “The UiO-66-NH₂ MSMPs process controllable shell numbers (1-4), tunable particle diameters (90-600 nm)...” It seems that only single-shell structure could be observed in 90 nm sized UiO-66-NH₂ samples. In other words, what is the minimum material size that can form a quadruple-shelled UiO-66-NH₂ structure?

Response: Thank you for the valuable comment. As mentioned in the introduction part, the average particle size of the prepared 4S-mesoUiO-66-NH₂ HoMSs is ~600 nm. The particle diameter of a mesoUiO-66-NH₂ HoMS is mainly determined by the thicknesses of each shell and the inter-shell spaces. Thus, by appropriately reducing the shell thicknesses and inter-shell spaces, the average particle size of the 4S-mesoUiO-66-NH₂ could be further decreased. As illustrated in Fig. 4b,c,j,k, the minimum sizes of 1S-mesoUiO-66-NH₂ and 2S-mesoUiO-66-NH₂ HoMSs are ~90 nm and ~170 nm, respectively. Based on this, we speculate that the smallest 4S-mesoUiO-66-NH₂ HoMS would have a size of $90 + (170-90) * 3 = 330$ nm.

Comment 3: According to the synthesis mechanism now proposed, how is the regulation of shell spacing and shell thickness realized?

Response: Thanks very much for the valuable comment. The shell thickness, interior cavity size, and inter-shell space of mesoporous UiO-66-NH₂ HoMSs can be precisely tailored at the nanoscale by tuning the nucleation kinetics of inhomogeneous dual-mesopore MOF precursors. During the formation of the single-layer MOF precursor particles, the nucleation kinetics is strongly associated with the concentration of acetic acid in the reaction system due to the competitive coordination between NH₂-BDC and acetic acid to [Zr₆O₄(OH)₄]¹²⁺ clusters. By increasing the concentration of acetic acid from 1.94 M to 3.01 M, the diameter of the single-layer MOF precursor particles increases from 90 to 360 nm (Fig. 4a,e). After etching, the shell thickness of the formed 1S-mesoUiO-66-NH₂ HoMSs can be varied from 30 to 60 nm, and the interior cavity size is tuned in the range from 30 to 240 nm (Fig. 4b-d,f-h). During the growth process of the second layer, the interfacial nucleation and growth kinetics of MOF layers is

greatly influenced by the amount of single-layer MOF cores. When using 80% and 20% of single-layer MOF particles prepared with the acetic acid concentration of 1.94 M in one pot as the cores (Fig. 4a), the diameter of the double-layer MOF precursor particles prepared in 2.50 M acetic acid solution increases from 170 to 280 nm (Fig. 4i,m). After etching, the thickness of the formed MOF layers increases from 40 to 55 nm (Fig. 4j-l,n-p). Accordingly, the space between the two shells of the resultant 2S-mesoUiO-66-NH₂ HoMSs varies from 25 to 75 nm after etching (Fig. 4j-l,n-p). The corresponding discussion has been added on Page 8 and 9, highlighted in yellow.

Figure 4 | Regulation of shell thickness, interior cavity size, and inter-shell space of mesoporous MOF particles. TEM images of (a,e) single-layer and (i,m) double-layer MOF precursor particles with different thicknesses of inhomogeneous MOF layers, TEM images of the corresponding products after etching treatment and schematic models: 1S-mesoUiO-66-NH₂ HoMSs prepared with the acetic acid concentrations of (b-d) 1.94 M and (f-h) 3.01 M, and 2S-mesoUiO-66-NH₂ HoMSs synthesized with (j-l) 80% and (n-p) 20% of single-layer MOF precursor particles prepared with the acetic acid concentration of 1.94 M in one pot as the cores.

Comment 4: As the author proposes this as a synthetic strategy, it is important to determine whether this strategy is universal. Can other MOF monomers be used to verify its effectiveness? What is the key prerequisite for realizing the dual-template-directed assembly?

Response: Thank you very much for the valuable comment. We have verified the universality of methodology and growth mechanism by successfully synthesizing hollow multi-shell mesoporous Hf-UiO-66-NH₂ and MOF-801 particles. Following the similar growth mechanism mentioned above, inhomogeneous mesoporous Hf-UiO-66-NH₂ and MOF-801 precursor particles could be formed by ODMB/F127-directed sequence assembly. By repeating such dual-template-directed assembly processes and subsequent selective etching, hollow double-shell mesoporous Hf-UiO-66-NH₂ and MOF-801 particles could be prepared (Supplementary Figs. 27-30). The corresponding

discussion has been added on Page 9, highlighted in yellow.

Supplementary Fig. 27 TEM images of (a,b) dual-mesopore core-shell Hf-UiO-66-NH₂ precursor particles prepared with both ODMB and F127 templates, (c,d) single-shell mesoporous Hf-UiO-66-NH₂ particles, (e,f) double-inhomogeneous-layer Hf-UiO-66-NH₂ precursor particles, and (g,h) double-shell mesoporous Hf-UiO-66-NH₂ particles.

Supplementary Fig. 28 XRD pattern of double-shell mesoporous Hf-UiO-66-NH₂ particles.

Supplementary Fig. 29 TEM images of (a,b) dual-mesopore core-shell MOF-801 precursor particles prepared with both ODMB and F127 templates, (c,d) single-shell mesoporous MOF-801 particles, (e,f) double-inhomogeneous-layer MOF-801 precursor particles, and (g,h) double-shell mesoporous MOF-801 particles.

Supplementary Fig. 30 XRD pattern of double-shell mesoporous MOF-801 particles.

Comment 5: Since several sized pores exist in the as-synthesized material, the roles of

the different sized pores should be clarified.

Response: Thank you for the valuable comment. Before the chemical etching, the MOF precursor particles contain three types of pores: the intrinsic micropores of the MOF itself, disordered smaller mesopores formed by ODMB-directed self-assembly, and radially cylindrical larger mesopores formed by F127-directed self-assembly. After the chemical etching, the defect-rich layers with disordered mesopores can be preferentially etched by acetic acid, leaving behind the intrinsic micropores of the MOF and the radially cylindrical mesopores guided by F127 in the resulting hollow multi-shell mesoporous MOF particles.

Comment 6: Related to question 4, can the enhancement mechanism of mass transfer process be verified with substrates containing different functional groups or substrates with different sizes?

Response: Thank you for the valuable comment. To demonstrate the difference in the mass diffusion between 3S-mesoUiO-66-NH₂ HoMSs and microporous MOF crystals, we further examine their performances in CO₂ cycloaddition reactions with two different epoxides propylene oxide and 1,2-epoxydodecane substituted with methyl and decyl groups, respectively. As expected, the 3S-mesoUiO-66-NH₂ catalyst shows higher conversion rate than microporous MOF catalyst (Supplementary Fig. 33). Distinct from the catalytic processes involving smaller substrate propylene oxide, the 3S-mesoUiO-66-NH₂ catalyst exhibits an almost twofold increase in the conversion rate of 1,2-epoxydodecane with larger substituted functional groups compared to the microporous MOF catalyst. This result suggests that mesoporous multi-shell nanostructures could enhance the mass transfer of larger molecules in the catalytic reactions, thereby improving their catalytic efficiency. The corresponding discussion has been added on Page 10, highlighted in yellow.

Supplementary Fig. 33 The conversion of propylene oxide and 1,2-epoxydodecane in the CO₂ cycloaddition reactions catalyzed by 3S-mesoUiO-66-NH₂ HoMSs and microporous MOF crystals.

Comment 7: What is the reason for the lack of catalytic performance data for quadruple-shelled UiO-66-NH₂ samples?

Response: Thank you for the valuable comment. In the CO₂ cycloaddition reaction

catalyzed by 4S-mesoUiO-66-NH₂ HoMSs, we achieved a conversion rate of ~96% after 10 h. Compared to 3S-mesoUiO-66-NH₂ HoMSs, the catalytic performance shows only a slight improvement, indicating that the conversion rate has reached its limit under such condition.

REVIEWER COMMENTS

Reviewer #1 (Remarks to the Author):

The quality of the manuscript has been significantly improved; thus I recommend the acceptance of this manuscript."

Reviewer #2 (Remarks to the Author):

In this revised manuscript, the authors have well addressed my concerns by point-by-point response and have improved their manuscript accordingly. Actually the desired catalyst should be the multi pore structure, such as macropore-mesopore-micropore from outside to inside. Author design the mesopore structure in outside of MOFs is very interestingly.

However, I have another question, the stability is a very important, especial in your case by using the metal sites as active center, I think the structure will easily collapse during catalysis. could you try using the nanoparticles (for example Pt, Au)as active center to construct the hollow multi-shell mesoporous metal-organic framework particles? If it can be achieved, I think it can be published in Nature Communications.

Reviewer #3 (Remarks to the Author):

The authors nicely addressed all my questions. I believe the manuscript is ready to publish.

Reviewer #4 (Remarks to the Author):

The authors have made revisions according to the comments. Still, some ambiguous points exist that weaken this work's impact.

1. The defect-rich thin pore wall in the templates should be further characterized. For example, the element distribution of precursors; and the chemical bonding between the surfactants, Zr-nodes, and ligands should be further clarified.
2. According to Fig. 3, the size of MOF precursor particles expands with the reaction time. Is there deconstructing and re-growth of UiO-66 crystals in the shell structures?
3. Besides, the values of inter-shell spacing of 4S-mesoUiO-66-NH₂ HoMSs and 5S-mesoUiO-66-NH₂ HoMSs are quite different in Fig 3. Which is possible to adjust them? See Adv. Mater. 2014, 26, 905.; Adv. Mater. 2012, 24, 1046. Adv. Mater. 2012,24,1046; Adv. Mater. 2012,24,1046; Angew. Chem. Int. Ed., 2021, 60, 6926.
4. The recent discovery of the formation of HoMS through precursor-rich templates has established a strong correlation with the periodic variation of precursor concentration on the template surface, referred to as concentration waves (Angew. Chem. Int. Ed. 2023, e202302621). In relation to question 2, if the assembly of UiO-66 crystals occurs during the etching process of the templates, it is plausible to explain the formation of mesoUiO-66-NH₂ HoMS using a similar theoretical framework. In this scenario, the precursors can be considered as being periodically distributed within the templates.

Response to the reviewers' comments

Reviewer #1:

The quality of the manuscript has been significantly improved; thus I recommend the acceptance of this manuscript."

Response: Many thanks to the reviewer for their useful comments and suggestions during the peer review process.

Reviewer #2:

In this revised manuscript, the authors have well addressed my concerns by point-by-point response and have improved their manuscript accordingly. Actually the desired catalyst should be the multi pore structure, such as macropore-mesopore-micropore from outside to inside. Author design the mesopore structure in outside of MOFs is very interestingly.

However, I have another question, the stability is a very important, especial in your case by using the metal sites as active center, I think the structure will easily collapse during catalysis. could you try using the nanoparticles (for example Pt, Au) as active center to construct the hollow multi-shell mesoporous metal-organic framework particles? If it can be achieved, I think it can be published in Nature Communications.

Response: Thank you very much for your valuable comment. We have introduced Au nanoparticles into 3S-mesoUiO-66-NH₂ HoMSs (denoting as Au@3S-mesoUiO-66-NH₂ HoMSs, Supplementary Fig. 37a,b). The obtained nanocomposites exhibit excellent activity and recycling stability in the hydrogenation of nitrobenzene (Supplementary Fig. 37c). The particle size of Au species slight increases and the nanostructure of 3S-mesoUiO-66-NH₂ HoMSs remains unchanged after five consecutive cycles (Supplementary Fig. 37d). The corresponding discussion has been added on Page 11 and Supplementary Fig. 37 Note, highlighted in yellow.

Supplementary Fig. 37 (a,b) TEM images of Au@3S-mesoUiO-66-NH₂ HoMSs, (c) the recycling tests of nitrobenzene hydrogenation over Au@3S-mesoUiO-66-NH₂ catalyst, and (d) TEM image of Au@3S-mesoUiO-66-NH₂ HoMSs after hydrogenation of nitrobenzene for five consecutive runs.

Reviewer #3:

The authors nicely addressed all my questions. I believe the manuscript is ready to publish.

Response: Many thanks to the reviewer for their useful comments and suggestions during the peer review process.

Reviewer #4:

The authors have made revisions according to the comments. Still, some ambiguous points exist that weaken this work's impact.

Comment 1: The defect-rich thin pore wall in the templates should be further characterized. For example, the element distribution of precursors; and the chemical bonding between the surfactants, Zr-nodes, and ligands should be further clarified.

Response: Thank you very much for your valuable comment. Since the ODMB surfactant and the organic ligand contain the same C, H, N, and O elements, the elemental distribution characterization may not provide additional information. Therefore, TEM and Fourier transform infrared spectroscopy (FTIR) analysis are used to further characterize the thin MOF pore wall. The defect-rich interior part of mesostructured MOF with disordered worm-like pores is formed by cooperative self-assembly of ODMB and MOF precursors at the beginning of the reaction (Supplementary Fig. 14a). The white domains, as indicated by yellow circles, are occupied by ODMB surfactants, while the black domains, marked by red circles, represent the defect-rich MOF pore wall (Supplementary Figure 14b). Fourier transform infrared spectroscopy (FTIR) analysis reveals the coordination interaction of the $[\text{Zr}_6\text{O}_4(\text{OH})_4]^{12+}$ clusters to the ligands and ODMB surfactants (Supplementary Fig. 14c). The band centered at 580 cm^{-1} is assigned to the characteristic Zr-(OC) asymmetric stretching. Besides, the strong absorptions at 2925 and 2850 cm^{-1} are caused by the C-H stretching vibration of methyl and methylene groups of ODMB. The corresponding discussion has been added on Page 7, highlighted in yellow.

Supplementary Fig. 14 (a,b) TEM images, (c) FTIR spectrum, and (d) XRD pattern of the UiO-66-NH₂ particles prepared with ODMB and F127 templates at 4 h.

Comment 2: According to Fig. 3, the size of MOF precursor particles expands with the

reaction time. Is there deconstructing and re-growth of UiO-66 crystals in the shell structures?

Response: Thank you very much for your valuable comment. As the growth times of inhomogeneous mesoporous MOF layers increases, there is no obvious sign that the mesostructure of the MOF precursor particles is deconstructed. Besides, the inner and intermediate shells of the MOF precursor particles with triple layers show the diameters of 140 and 270 nm which are the same as the diameters of single-layer MOF precursor particles and double-layer MOF precursor particles, respectively. This indicates that the MOF precursor does not undergo subsequent regrowth. Based on the observation mentioned above, we conclude that there are no deconstructing and regrowth processes in the MOF shell during the growth process.

Comment 3: Besides, the values of inter-shell spacing of 4S-mesoUiO-66-NH₂ HoMSs and 5S-mesoUiO-66-NH₂ HoMSs are quite different in Fig 3. Which is possible to adjust them? See Adv. Mater. 2014, 26, 905.; Adv. Mater. 2012, 24, 1046. Adv. Mater. 2012,24,1046; Adv. Mater. 2012,24,1046; Angew. Chem. Int. Ed., 2021, 60, 6926.

Response: Thank you very much for your valuable comment. The inter-shell spacing of the mesoporous UiO-66-NH₂ HoMSs can be tuned by regulating the nucleation kinetics of inhomogeneous dual-mesopore MOF precursors, as elucidated in Paragraph 6 within the results and discussion section. It is observed that the inter-shell spacing of 5S-mesoUiO-66-NH₂ HoMSs is slightly larger compared to the other samples shown in Fig 3, because it becomes increasingly difficult to precisely manage the kinetics of nucleation and growth, particularly when extending the growth times of these inhomogeneous mesoporous MOF layers. We have cited the related studies you provided as Ref. 27, 28, and 29.

Comment 4: The recent discovery of the formation of HoMS through precursor-rich templates has established a strong correlation with the periodic variation of precursor concentration on the template surface, referred to as concentration waves (Angew. Chem. Int. Ed. 2023, e202302621). In relation to question 2, if the assembly of UiO-66 crystals occurs during the etching process of the templates, it is plausible to explain the formation of mesoUiO-66-NH₂ HoMS using a similar theoretical framework. In this scenario, the precursors can be considered as being periodically distributed within the templates.

Response: Thank you very much for your valuable comment. The literature reported the precursors inside the template become enriched on the surface and form shell structures during the template shrinking. The periodical variations of the precursor provide more guiding information for fabricating HoMS (Angew. Chem. Int. Ed. 2023, e202302621). On the other hand, our synthetic methodology depends on the directly construction of MOF precursor particles with two different alternately distributed mesostructured MOFs layers. After selective etching, the multi-shell hollow MOF particles are generated. The related work has been cited as Ref. 26.

REVIEWERS' COMMENTS

Reviewer #2 (Remarks to the Author):

Authors have answered all questions and it should be published on the Nature communications.

Reviewer #4 (Remarks to the Author):

The authors have addressed all the questions. I recommend to accepting this manuscript.